# High Glucose Levels Promote Switch to Synthetic Vascular Smooth Muscle Cells via Lactate/GPR81

**DOI:** 10.3390/cells13030236

**Published:** 2024-01-26

**Authors:** Jing Yang, Glenn R. Gourley, Adam Gilbertsen, Chi Chen, Lei Wang, Karen Smith, Marion Namenwirth, Libang Yang

**Affiliations:** 1Hubei Key Laboratory of Embryonic Stem Cell Research, Hubei University of Medicine, Shiyan 442000, China; 2Institute of Virology, Hubei University of Medicine, Shiyan 442000, China; 3Department of Infectious Diseases, Renmin Hospital, School of Basic Medical Sciences, Hubei University of Medicine, Shiyan 442000, China; 4Department of Pediatrics, University of Minnesota, Minneapolis, MN 55455, USA; gourleyg@umn.edu (G.R.G.); namenwir@umn.edu (M.N.); 5Department of Medicine, University of Minnesota Medical School, Minneapolis, MN 55455, USA; gilbe398@umn.edu (A.G.); smith493@umn.edu (K.S.); 6Department of Food Science and Nutrition, CFANS, University of Minnesota, St Paul, MN 55108, USA; chichen@umn.edu (C.C.); wang2232@umn.edu (L.W.)

**Keywords:** smooth muscle cells, phenotype change, glucose, lactate, GPR81, diabetes, vascular complication, TallyHo mouse

## Abstract

Hyperglycemia, lipotoxicity, and insulin resistance are known to increase the secretion of extracellular matrix from cardiac fibroblasts as well as the activation of paracrine signaling from cardiomyocytes, immune cells, and vascular cells, which release fibroblast-activating mediators. However, their influences on vascular smooth muscle cells (vSMCs) have not been well examined. This study aimed to investigate whether contractile vascular vSMCs could develop a more synthetic phenotype in response to hyperglycemia. The results showed that contractile and synthetic vSMCs consumed high glucose in different ways. Lactate/GPR81 promotes the synthetic phenotype in vSMCs in response to high glucose levels. The stimulation of high glucose was associated with a significant increase in fibroblast-like features: synthetic vSMC marker expression, collagen 1 production, proliferation, and migration. GPR81 expression is higher in blood vessels in diabetic patients and in the high-glucose, high-lipid diet mouse. The results demonstrate that vSMCs assume a more synthetic phenotype when cultured in the presence of high glucose and, consequently, that the high glucose could trigger a vSMC-dependent cardiovascular disease mechanism in diabetes via lactate/GPR81.

## 1. Introduction

Type 2 diabetes is diagnosed as advanced stage when vascular complications have already occurred. Macro- and microvascular diabetic complications are mainly due to prolonged exposure to hyperglycemia in addition to other risk factors such as arterial hypertension, dyslipidemia, and genetic susceptibility [1,2]. Diabetes vascular diseases involve many organs, endothelial cells, pericytes, astrocytes, smooth muscle cells, and inflammatory factors [1,2,3,4,5].

Vascular smooth muscle cells (vSMCs) play a complicated role in diabetes. Although vSMCs are perhaps most frequently studied for their role in arterial contraction, they are found in a wide range of organs and participate in a variety of other physiological activities and repair mechanisms [6]. The diversity of vSMC functions is reflected in their phenotypes, which compose a spectrum that ranges from predominantly contractile cells at one end to predominantly synthetic cells at the other end. The contractile and synthetic phenotypes are characterized by substantial differences in marker expression, morphology, and activity [7,8]. Proliferating vSMCs are a major source of extracellular matrix in atherosclerotic plaque and in the fibrous cap covering the plaque [9]. Smooth muscle cells proliferate, accumulate, and show phenotype switch in diabetes-accelerated atherosclerotic lesions but no direct growth promotion from high glucose [2,10,11]. vSMC proliferation in response to hyperglycemia is an important process in the development of arterial vessel hyperplasia in diabetes [12,13]. The mechanisms by which hyperglycemia activates a phenotypic switch in vSMCs, however, are poorly understood.

Elevated lactate levels in diabetic patients have been reported often, but the relational mechanism between lactate levels and diabetic complications remains unclear [14,15,16]. Lactate is a substrate for gluconeogenesis and an energy source for cardiac and type I skeletal muscle fibers; however, relatively recent reports indicate that lactate could be taken up by endothelial cells and human cytotoxic T lymphocytes [17,18], inhibits phosphofructokinase [19], alters gene expression in L6 muscle cells [20], participates in T-cell migration [21], and contributes to tumor growth [22,23,24,25]. G protein-coupled receptor 81 (GPR81) is a protein with seven transmembrane domains and transduces extracellular signals through heterotrimeric G proteins. It has been identified as a lactate receptor and is responsible for cancer cell migration and invasion [24,26]. Even under fully oxygenated conditions, vSMCs produce a substantial amount of lactate [27], and both local and systemic lactate concentrations increase in response to regional ischemia and cardiac arrest, as well as shock, burns, trauma, and other conditions [28,29]. Previously, we reported that lactate promotes a smooth muscle cell switch to the synthetic phenotype in a hypoxic environment [30]. We hypothesize that high glucose promotes vSMC proliferation and ECM production via lactate/GPR81 rather than glucose directly. Here, we show that high glucose concentrations increased lactate production and GPR81 expression in vSMCs. Additionally, we find that increased lactate/GPR81 promotes the synthetic phenotype of vSMCs and, consequently, stimulates synthetic vSMC-dependent mechanisms of proliferation in response to high glucose.

## 2. Materials and Methods

### 2.1. Maintenance and Differentiation of Human Induced Pluripotent Stem Cells

Human induced pluripotent stem cells (hiPSCs) were maintained on vitronectin-coated plates in DMEM-F12 (Life Science, St Louis, MO, USA) supplemented with 20% knockout serum replacement (Invitrogen, Waltham, MA, USA), 0.1 mmol/L MEM nonessential amino acid solution, 2 mmol/L L glutamine, 0.1 mmol/L β-mercaptoethanol (Sigma, Tulsa, OK, USA), and 4 ng/mL basic fibroblast growth factor (Thermo Fisher Scientific, Waltham, MA, USA). Passage 10 ≈ 15 of hiPSC was used in this project. hiPSCs were differentiated into hiPSC-vSMCs as described previously [7,30]. The standard culture medium for vSMCs consisted of Roswell Park Memorial Institute medium (RPMI)-1640 with 1 × B27 minus insulin and a growth factor mixture (human basic fibroblast growth factor, 2 ng/mL; human epidermal growth factor, 0.5 ng/mL). The medium for vcSMCs consisted of glucose-free RPMI-1640 with 0.2 × B27 minus insulin. Glucose and L-lactate are added as needed in the experimental design.

### 2.2. Human Tissue

Diabetic patient samples were used to present GPR81 expression change in blood vessels. Tissue specimens were obtained through the LTCDS (Liver Tissue Cell Distribution System, NIH Contract #HHSN276201200017C). Tissue samples: tissue diced and tissue sections from 8 diabetic livers, 5 diabetic lungs, 8 control livers (1 normal, 2 alcoholic fibrosis, 2 acute hepatitis), and 3 lungs (2 lung fibrosis, 2 emphysema, 1 normal region of bronchocarcinoma) (gender and age-matched age ≤ 5 years apart) were provided by UMN tissue bank and delivered on dry ice or cut by requirement. The IRB Human Subjects Committee of the University of Minnesota, study Number: 1305M32841, approved this study.

### 2.3. Flow Cytometry Analysis

vSMCs with more than 95% positive markers were used in experiments. Flow cytometry sorting was used to identify and collect vSMCs as described before [7,30].

### 2.4. Glucose Uptake Assay

Glucose and lactate kits (Abcam, Cambridge, MA, USA) were used in the assay. The cells were cultured in RPMI-1640 with B27 minus (or serum-free) for 18 h and the cells were washed with PBS x2. The cultures were then incubated for 30 min in basic RPMI-1640 (no glucose, Gibco, Grand Island, NY, USA) with KRPH for 40 min. Then, the cells were cultured with 2DG for 20 min. After following the manufacturer’s instructions, the 2DG6P was measured at 420 nm with a SpectraMax M3 microplate reader (Molecular Devices, Silicon Valley, USA).

### 2.5. Glucose Consumption and Lactate Turnover Assay

The cells (2 × 10^4^) were seeded into 48-well plates, incubated in RPMI-1640 with 1 × B27 minus insulin for 24 h, and then changed in RPMI-1640 without B27 for 24 h; then, the cells were lysed, and the lysate and medium were collected. Glucose and lactate levels at different time points were determined with glucose assay (Sigma) and lactate assay (Biovision, Milpitas, CA, USA) kits per the manufacturer’s instructions; fluorescent and colorimetric densities were evaluated at 570 nm with a Biotek Dynax2.

### 2.6. ATP Concentration Measurement and Lactate Concentration Measurement

ATP and lactate measurements were conducted as described before [30]. Measurements were quantified with a SpectraMax M3 microplate reader (Molecular Devices). Glucose uptake assay, glucose consumption, lactate turnover assay, lactate concentration measurement, and ATP measurement were used in glucose metabolism analysis in vSMCs.

### 2.7. ELISA, Proliferation, and Apoptosis

Collagen I and lactate-dehydrogenase levels were measured with an ELISA kit (R&D systems, USA or Life Span biological, Seattle, DC, USA); cell proliferation and apoptosis were measured with MTT proliferation and apoptosis kits (Roche. Indianapolis, IN, USA). Measurements were quantified with a SpectraMax M3 microplate reader (Molecular Devices).

### 2.8. Migration Assay

Migration was evaluated across 8 µm pore size membranes in 96-well tissue culture plates with a cell migration kit (Millipore, Burlington, MA, USA). Cells were cultured in RPMI-1640 B27 for 24 h, trypsinized, and then added with 150 µL of RPMI-1640 B27 to the upper chamber (1 × 10^5^ cells per chamber). The lower chamber contained 500 µL of RPMI-1640 B27 and was coated with (positive control) or without (negative control) Matrigel. The cells were incubated for 12 h at 37 °C, and the cells that had migrated to the lower chamber were detected with CyQuant GR Dye.

### 2.9. GLUT1, LDHa, and GPR81 Knockdown

Cells were transfected with *GLUT1* (NM 006516), *LDHA* (NM 005566), and *GPR81* (NM 032554) shRNA or scramble shRNAs (IDT Biotechnology, Coralville, IA, USA, and Genomics Center, UMN) by using Fu-Gene 6 (Promega, Madison, WI, USA) per the manufacturer’s instructions. Briefly, the shRNAs GIPZ and reagent (in a 1:2 ratio) were dissolved in RPMI-1640 and incubated for 30 min at room temperature; then, the shRNA-reagent complexes were added to the cells, and the cells were incubated at 37 °C. After 4 h, the transfection medium was replaced with fresh medium and the cells were cultured for 2 more days before use in subsequent experiments. Transduction with *GPR81* shRNA or control RNA particles was performed in RPMI-1640 with 2 μg/mL Polybrene (Millipore); then, the cells were incubated overnight, the transduction medium was replaced with fresh medium, and the cells were cultured for 2 days before use in subsequent experiments.

### 2.10. Real-Time Reverse Transcription PCR

Real-time RT-PCR analysis was performed as described before [30]. The primer sequences are listed in Table 1 below.

### 2.11. Western Blot Analysis

Western blot analysis was conducted as described before [30]. The protein bands were visualized with ECL Plus per the manufacturer’s instructions and saved with Chemidoc it imager (Upland, CA, USA).

### 2.12. Immunocytochemistry

Immunocytochemistry analysis was carried out as described before [30]. Positively stained cells were quantified on serial sections using Image J software, Version 1.5.

### 2.13. Mouse TallyHo Model of High-Lipid, High-Glucose Diet

TallyHo C57BL/6J mice, used in diabetes mouse models, were used in this project (Jackson Laboratory, Bar Harbor, ME, USA). All experiments were carried out according to a protocol approved by the Institutional Committee for Use of Animals in Research of the Hubei University of Medicine. The mice treatment was modified from a published method [31]. The two groups of six animals each were fed for 12 weeks as follows: control, mice were fed a low-fat diet and drank tap water; HGHL, mice were fed a high-fat diet and tap water with 10% glucose. Food and drink were changed daily. Mouse chow was made by the animal center and contained 6% chicken-derived fat, 40.7% carbohydrates, and 24% protein. The HGHL was prepared by adding 22% chicken-derived fat.

### 2.14. Metabolomics Analysis

The LC-MS-based metabolomic analysis comprised sample preparation, chemical derivatization, LC-MS analysis, data deconvolution and processing, multivariate data analysis (MDA), and marker characterization and quantification [32]. A total of 3 × 10^6^ vSMCs at 70% confluency as designed groups were washed 3X, cultured with designed glucose concentration (containing 5 µCi/mL ^13^C-glucose), and lysed, as metabolomics required, at 30 min and 24 h. For detecting the metabolites containing functional groups in their structures, the samples were derivatized with dansyl chloride (DC) prior to the LC-MS analysis. Briefly, 5 μL of the sample or standard was mixed with 5 μL of 100 μmol/L *p*-chlorophenylalanine (internal standard), 50 μL of 10 mmol/L sodium carbonate, and 100 μL of DC solution (3 mg/mL in acetone). The mixture was incubated at 25 °C for 15 min and centrifuged at 18,000× *g* for 10 min, and the supernatant was transferred into an HPLC vial for LC-MS analysis. For detecting carboxylic acids, aldehydes, and ketones, the samples were derivatized with HQ prior to the LC-MS analysis. Briefly, 2 μL of sample was added into 100 μL of freshly prepared acetonitrile solution containing 1 mmol/L DPDS, 1 mmol/L TPP, and 1 mmol/L 2-hydrazinoquinoline (HQ). The reaction mixture was incubated at 60 °C for 30 min, chilled on ice, and then mixed with 100 μL of ice-cold H_2_O. After centrifugation at 18,000× *g* for 10 min, the supernatant was transferred into an HPLC vial for LC-MS analysis.

### 2.15. Proteomics Analysis

A total of 5 × 10^6^ vSMCs at 75~90% confluency as designed groups were washed 3X and lysed as proteomics required. Protein concentrations were determined in desalted samples with Bradford reagent (Bio-Rad, Hercules, CA, USA), and then samples containing equal amounts of protein (20 µg) were labeled with iTRAQ reagent (Applied Biosystems, Foster City, CA, USA) per the manufacturer’s instructions, as described previously [30]. Assessments were performed in triplicate with iTRAQ 8-plex kits.

Strong cation exchange chromatography, LC-MALDI (liquid chromatography–matrix-assisted laser desorption/ionization), 4800 MS/MS, and peptide and protein identification and isolation were conducted as described previously [30].

### 2.16. Ingenuity Pathway Analysis

The selected proteins were imported to the Ingenuity Pathway Analysis software (http://www.ingenuity.com, 12 May 2023) to identify their associated pathways, biological functions, and diseases. The number of genes associated with each biological function or disease was counted, and *p* values were calculated via the Fisher exact test.

### 2.17. Statistical Analysis

All statistical analyses were performed with the Statistical Package for Social Sciences for Windows version 17 software (SPSS, Chicago, IL, USA). Values are presented as mean ± SD, and significance was evaluated via the Student *t* test. A *p* value of <0.05 was considered statistically significant.

## 3. Results

### 3.1. Contractile vSMCs and Synthetic vSMCs Respond to High Glucose Concentrations Differently in Cell Culture

To observe if vSMC subtypes respond differently to high glucose concentration in cell culture, contractile and synthetic vSMCs derived from hiPSCs were used for in vitro studies. vSMCs were cultured in a high-glucose-concentration culture. Measurements of glucose uptake (Figure 1a), ATP production (Figure 1b), and cell viability (Figure 1c) in vSMCs were significantly different between contractile vSMCs and synthetic vSMCs. Glucose uptake measurements were significantly greater in synthetic vSMCs (almost three times) than in human arterial SMCs and in contractile vSMCs. ATP levels (Figure 1b) were 30% higher in synthetic vSMCs than in contractile vSMCs. Synthetic vSMCs grow much faster than contractile vSMCs (Figure 1c). We studied lactate production in both types of vSMC with cells cultured in normal (5 mM) (Figure 1d) and high (25 mM) (Figure 1e) glucose concentrations. Lactate production (Figure 1d) in contractile vSMCs was lower than in synthetic vSMCs in normal-glucose medium but variable depending upon incubation time in high-glucose medium (Figure 1e). When we measured lactate in the medium of vSMCs cultured in normal or high glucose concentrations for 16 h (Figure 1f), there was no significant difference between contractile vSMCs and synthetic vSMCs in normal-glucose culture. But in high glucose concentrations, lactate levels in the contractile vSMCs medium were 62% higher than in the synthetic vSMC medium. In cell lysates, lactate levels in contractile vSMCs were lower than in synthetic vSMCs in normal-glucose culture, but in high-glucose culture, lactate levels in contractile vSMCs were 30% higher than in synthetic vSMCs (Figure 1g). These findings suggest that there are differences in glucose and lactate metabolism between contractile vSMCs and synthetic vSMCs in high-glucose cultures. Contractile vSMCs release more lactate into the medium than synthetic vSMCs in high-glucose culture, but synthetic vSMCs had more lactate in cell lysate than contractile vSMCs.

### 3.2. Contractile vSMCs and Synthetic vSMCs Have Different Metabolism in High-Glucose-Concentration Culture

We then aimed to validate the differences in glucose metabolism between contractile vascular smooth muscle cells (vSMCs) and synthetic vSMCs using proteomics and metabolomics methods. Our analysis of synthetic vSMCs revealed lower expression of proteins and metabolites involved in the glycolysis pathways and higher expression of enzymes involved in lactate production and the tricarboxylic acid (TCA) cycle (Figure 2a). In terms of glycolysis, we found that proteins such as PFKc, ALDOC, TPI1, PGK1, PGAM1, and ENO1, along with metabolites like ^13^C6-G1P, ^13^C6-F6P, and ^13^C3-GA3P, were significantly lower in synthetic vSMCs compared to contractile vSMCs. Synthetic vSMCs displayed higher levels of pentose phosphate pathway proteins 6GPD and 6PGL, as well as metabolites ^13^C6-6PG and ^13^C6-RUSP. Additionally, synthetic vSMCs exhibited increased lactate production. In the TCA cycle (Figure 2a), proteins such as PDHe1, DLAT, SDH, and MDH, along with metabolites ^13^C3-Acetyl CoA, ^13^C2-citrate, and ^13^C2-isocitrate, were higher in synthetic vSMCs compared to contractile vSMCs. These findings suggest that synthetic vSMCs utilize the TCA cycle more effectively to produce ATP and promote cell growth.

These proteomics and metabolomics analyses underscore dynamic differences in the metabolism of glucose and lactate between these cell types (Figure 2). Specifically, when comparing contractile vSMCs to synthetic vSMCs, we observed that many glycolytic enzymes and metabolites were elevated in contractile vSMCs, while lactate production was reduced in contractile vSMCs. Our proteomics analysis indicated higher expression of glucose transporter 1 (GLUT1), LDHa, and monocarboxylate transporter 1 (MCT1) in synthetic vSMCs than in contractile vSMCs. Western blot and RT-PCR analysis showed that synthetic vSMCs expressed more GLUT1 and LDHa than contractile vSMCs in cultures with high glucose concentration (Figure 2b). Both RT-PCR and Western blot analyses indicated that GLUT1 and LDHa levels were higher in cultures with high glucose in synthetic vSMCs compared to a normal-glucose culture. However, there was no significant change in contractile vSMCs between cultures with normal glucose and high glucose. When we knocked down *GLUT1* with *GLUT1* shRNAs in vSMCs, ATP production and cell proliferation in contractile vSMCs remained unchanged, but they decreased in synthetic vSMCs in high-glucose culture (Figure 2c). Knocking down *LDHa* with *LDHa* shRNAs in vSMCs yielded similar results to *GLUT1* knockdown (Figure 2d). Western blot analyses showed inhibition of LDHa and GLUT1 also reduced the expression of MCT1, GLUT1, and ki67. Levels of MCT1, GLUT1, and ki67 were reduced when vSMCs were transduced with *GLUT1* shRNA and *LDHa* shRNA (Figure 2e). Consequently, GLUT1 and LDHa contribute to the differences between contractile vSMCs and synthetic vSMCs in cultures with high glucose concentrations. These findings suggest that LDHa and GLUT1 both play critical roles in ATP production and cell proliferation in vSMCs in high-glucose culture. In summary, these data demonstrate that there are increased levels of many glycolytic enzymes and metabolites in contractile vSMCs, while there are higher levels of lactate and TCA enzymes and metabolites in synthetic vSMCs, suggesting that synthetic vSMCs make better use of the TCA cycle to produce ATP and promote cell growth.

### 3.3. High Glucose Concentration Promotes vSMC Phenotype Switch

We next sought to determine if high glucose concentration affects the vSMC phenotype. When contractile vSMCs were cultured on Matrigel for a longer period (4 days) in a high-glucose-concentration medium, contractile vSMCs tended to become less spindle-shaped and develop a more irregular morphology associated with synthetic vSMCs (Figure 3a). High-glucose-cultured cells were significantly more proliferative than normal-glucose-cultured cells when evaluated via both the MTT assay and ki67 staining. Contractile vSMCs proliferate 21% more in high-glucose cultures than in normal-glucose cultures (Figure 3b). Immunofluorescent stain showed that high-glucose-treated cells had 11% more ki67-positive cells than untreated cells; ki67+ cells were significantly increased in high-glucose culture while not increased in normal-glucose culture (Figure 3c). Measurements of cell motility were also 49% greater when the cells were cultured in high glucose than in a normal-glucose medium (Figure 3d). Analysis of extracellular matrix (ECM) and vSMC markers with both RT-PCR and Western Blot revealed the cell phenotype shift to synthetic vSMCs. Expression of collagen I and vimentin (VMT) were increased, both markers for synthetic vSMCs, and MYH11 and transgelin (TAGN), both markers for contractile vSMCs, were decreased, while ACTA2, a marker for vSMCs, did not change significantly (Figure 3e,f). In our observation of immunocytometry, there was more vimentin expressed in high-glucose-cultured vSMCs compared to that seen in normal-glucose-cultured vSMCs (Figure 3g). In summary, the high-glucose-treated vSMCs led to a lower expression of MYH11 and TAGN but higher proliferation and migration and produced more ECM proteins, which match with characteristics of the synthetic vSMC phenotype. Thus, a high glucose concentration may induce cultured vSMCs to assume a predominantly synthetic phenotype.

### 3.4. High Glucose Concentration Promotes vSMC Synthetic Phenotype via Lactate-GPR81

Lactate has been shown to promote vSMC phenotype switch [30]. GPR81 is a lactate receptor and is responsible for cancer cell migration and invasion [20,26]. We hypothesized that GPR81 plays a similar role in vSMCs in high-glucose culture since high glucose leads to high lactate production. GPR81 expression was investigated and the *GPR81* gene was manipulated to observe its effect on vSMC function. GPR81 in contractile vSMCs in high-glucose treatment and lactate treatment was significantly higher than the normal culture in RT-PCR and Western blot analysis (Figure 4a,b). When we knocked down GPR81 with GPR81shRNA, the proliferation and migration of vSMCs in high-glucose culture were inhibited (Figure 4c–g). GPR81 shRNA vSMCs had lower cell proliferation than the scramble shRNA vSMCs. They also expressed lower proliferation marker ki67 (Figure 4e). When we investigated the motility of vSMCs, GPR81 shRNA vSMCs had lower cell migration than the scramble shRNA vSMCs; they also express lower matrix metalloproteinase-2, and its expression level goes along with cell migration (MMP2) (Figure 4f,g). Knocking down *GPR81* reversed changes in the expression of contractile and synthetic vSMC markers in high-glucose treatment. The contractile vSMC marker *MYH11* was decreased in RT-PCR and in Western blot (Figure 4h), but the changes were not observed in *GPR81*shRNA-transduced contractile vSMCs. When we observed contractile and synthetic markers, synthetic SMC markers vimentin and collagen I were increased; contractile markers *TAGN* and *MHY11* were decreased in RT-PCR, Western blot, and ELISA, but the increase was not observed in *GPR81*shRNA-transduced contractile vSMCs (Figure 4h–j). These results suggest high glucose promotes the synthetic vSMC phenotype through lactate receptor GPR81. Taken together, these data support the concept that lactate/GPR81 is a key axis in synthetic vSMC development.

### 3.5. GPR81 Regulates vSMC via PGC1a, MCTs, and CD147

Synthetic vSMCs have higher proliferation, higher mobility, and produce more ECM proteins than contractile vSMCs [30,33]. PGC-1α is a transcriptional coactivator that regulates the genes involved in energy metabolism and is the master regulator of mitochondrial biogenesis. This protein interacts with the nuclear receptor PPAR-γ, which permits the interaction of this protein with multiple transcription factors [34]. Extracellular matrix metalloproteinase inducer, CD147 or basigin, is a widely distributed cell surface glycoprotein that is involved in cell migration and invasion. MCTs are responsible for lactate transportation [35]. High-glucose and lactate treatment stimulates PGC1a, MCT, and CD147 expression. RT-PCR analysis demonstrates that *PGC1a*,* MCT1*,* MCT4*, and *CD147* were increased in high-glucose and lactate treatment (Figure 5a). Western blot analysis showed similar changes in these samples (Figure 5b). To determine how GPR81 affects vSMCs, PGC1α, MCT, and CD147 levels were observed in GPR81 knocked down vSMCs. Knocking down GPR81 with shRNA in contractile vSMCs inhibited the PGC1α, MCTs, and CD147 in high-glucose treatment. MCT1, MCT4, PGC-1a, and CD147 were all increased in scramble shRNA-transduced vSMCs in RT-PCR but not in *GPR81* shRNA-transduced vSMCs in high-glucose culture (Figure 5c). Western blot showed a similar pattern. Expression of MCT1, MCT4, PGC1a, and CD147 in scramble shRNA-transduced vSMCs was increased in high-glucose culture, but their levels were inhibited in GPR81 shRNA-transduced vSMCs (Figure 5d). These data support the idea that GPR81 plays a role in the vSMC phenotype switch via PGC1α, MCTs, and CD147.

### 3.6. GPR81 Expression in vSMCs in Human Tissue and the Diabetic Mouse Model

To investigate if GPR81 and vSMCs change in diabetes in vivo, GPR81 expression and smooth muscle markers were observed in diabetic human tissues and the mouse model. GPR81 expression and smooth muscle markers were observed in diabetic tissues. Tissues in diabetic patients and in the control group were frozen with liquid nitrogen and stored at −80 °C for lactate biochemistry analysis or fixed with standardized clinical paraffin for tissue slides for IHC analysis. GPR81 expression and SMA protein expression are increased significantly in diabetic tissues compared to the control group. Of the related proteins studied, LDHa mRNA and protein levels are increased in diabetic lung tissue. GPR81 and SMA were increased in those tissues too. There are more blood vessels in the diabetic patient tissue based on SMA stain in IHC using an image J program (Figure 6a–e). There are two times more blood vessels in diabetic tissue than in nondiabetic tissue identified through image density quantification in both the blood vessel and GPR81-positive strains. LDHa activity and lactate were measured in frozen lung tissue samples; diabetic tissue had higher LDHa activity and higher lactate concentration than the control samples (Figure 6f). Diabetes tissue had 60% more LDHa activity than the control tissues.

Similar patterns were also observed in the mouse model. TallyHo mice were fed a diet that was either normal or high in sugar and lipids (HGHL). After 16 weeks of this diet, the tissues were then analyzed with immunohistochemistry for SMA and GPR81 and biochemistry analysis for LDHa. Blood vessel and GPR81-positive vSMCs are increased in high-glucose, high-lipid diet TallyHo mice compared to control groups in cardiac tissues and kidney tissues of TallyHo mice (Figure 6g,h). There were more GPR81-positive blood vessel cells in HGHL mice than in the control tissue sections (Figure 6i). These findings suggest that high glucose could promote blood vessel cell proliferation and GPR81 expression in blood vessels. In the related protein studied, LDHa activity and lactate level were increased by more than 50% in diabetic kidney tissue than in the control tissue (Figure 6j). These findings imply that there is more proliferation in synthetic vSMCs, and GPR81 is expressed higher in vSMCs in TallyHo HGHL mice.

## 4. Discussion

Type 2 diabetes has numerous impacts on the body, including the heart, blood vessels, nerves, eyes, and kidneys. Diabetes vascular diseases involve many organs and endothelial cells, pericytes, astrocytes, smooth muscle cells, and many inflammatory factors [1,2,3,4,5]. But the exact mechanism of vascular complications is still not clearly described. vSMCs have been considered a key player in many steps in vascular disease formation [6,9]. vSMCs contribute to many different plaque cell phenotypes, including extracellular matrix-producing cells of the fibrous cap, macrophage-like cells, foam cells, mesenchymal stem cell-like cells, and osteochondrogenic cells [8,36]. vSMCs proliferate, accumulate, and show phenotype switch in diabetes-accelerated atherosclerotic lesions but no direct growth promotion from high glucose [2,10,11]. vSMCs do not terminally differentiate but retain remarkable plasticity. In response to vascular injury or alterations in local environmental cues, differentiated/contractile vSMCs can switch to more synthetic vSMCs [33].

Glucose metabolism is abnormal in diabetes [37,38] in that cellular uptake and use of glucose is variable depending on the cell type. Our data show that high glucose concentrations result in increased lactate release in both contractile and synthetic vSMCs. Previous research has shown that synthetic vSMCs can take up more lactate than contractile vSMCs [30]. The current findings show that synthetic vSMCs can use lactate better than contractile vSMCs. It is well known that vSMCs can produce lactate even in an oxygen-rich environment, which is similar to the “Warburg effect” in cancer cells. Cancer cells depend on glycolytic phenotypes characterized by high rates of glucose uptake and lactate and ATP production, regardless of oxygen concentration, which benefits cancer cell survival [39]. Thus, cancer cells can use both glucose and lactate, a benefit when one cell subgroup uses glucose and releases lactate and another cell subgroup uses lactate. Our proteomic and metabolomic analyses show that synthetic vSMCs can use lactate as an energy resource. In fact, there are some reports that suggest that endothelial cells take up glucose and release lactate [18,38], forming a high-lactate cue for vSMCs. Then, high glucose results in a high-lactate cue where vSMCs can both generate and utilize lactate in their metabolism.

Lactate is related to some pathological changes in diabetes and other diseases but its mechanism needs further study [12,38,40]. GPR81 is a G*_i_*-coupled receptor, which is expressed mainly in adipocytes but is also present at low levels in a variety of normal cells including in the liver, kidney, and heart [41]. Lactate and GPR81 levels are related to changes in cell function in immune cells and cancer cells [26,42,43,44]. It is also associated with cardiovascular pressure regulation [43]. In this study, different phenotypes of vSMCs express GPR81 differently depending on the glucose concentration. High glucose levels stimulated vSMCs to express increased GPR81 in synthetic vSMCs. In contrast, contractile vSMCs express low levels of GPR81. This variable expression pattern of GPR81 suggests a potentially broad role of GPR81 with vSMC function. Our functional studies indicated that GPR81 is important for lactate regulation of proteins involved in lactate uptake and metabolism. GPR81 is critical for ECM protein production of synthetic vSMCs. Moreover, GPR81 levels correlated with rates of vSMC proliferation and migration. These data suggest that the expression of GPR81 is fundamental for the vSMC subtype switch and synthetic vSMC function. MCTs, PGC-1α, and CD147 have been shown to be related to cell proliferation and ECM production [18,45,46,47,48,49,50]. The expression of MCTs involved in lactate transport has previously been shown to be regulated by a variety of factors, including oxygen levels via HIF-1, cytokines, p53, and PGC-1α [35]. Our data demonstrate that lactate and PGR81 influence the expression of LDHa, MCTs, PGC-1α, and CD147. This further suggests that GPR81 is necessary for the vSMC subtype switch and synthetic vSMC function.

Diabetes-related vascular complications are intricate and can damage the heart, eye, brain, kidney, lungs, skin, and many other organs [3,5,51]. vSMCs in vascular disease characterize the identity of different vSMC regions by both transcriptional and epigenetic mechanisms to determine which developmental signatures are preserved in the adult vasculature [8,33,36]. There are many different abnormal pathological changes described [36,52]. We previously demonstrated the vSMC phenotype switch and related lactate change [7,30]. Our current research demonstrates alterations in GPR81 expression in tissue and blood vessel characteristics in diabetic patients and a diabetic mouse model. We investigated vascular alterations using the TallyHo mouse, a model used in type 2 diabetes research [53,54]. Our data showed that GPR81 levels change in the heart and kidney in mice and lung tissue in humans. Since GPR81 lacks an antagonist [55], our data only showed vascular proliferation but not a proper interruption for it. Lactate/GPR81 is then a potential therapeutic target for diabetic vascular complications.

## 5. Conclusions

We have shown that high glucose concentration promotes the synthetic vSMC phenotype via lactate/GPR81. GPR81 is highly expressed in synthetic vSMCs and high glucose levels promote the switch to the synthetic vSMC phenotype. GPR81 is critical for sensing extracellular lactate and synthetic vSMC functions such as proliferation, migration, and ECM production. Activation of GPR81 by lactate leads to increased expression of MCTs, CD147, and PGC-1α, which are critical for lactate transport and metabolism. These data demonstrate, for the first time, the presence of GPR81 in diabetic tissue and vSMCs cultured in a high-glucose-concentration medium. Further, we have demonstrated the role of GPR81 in the TallyHo mouse model. The data suggest that GPR81 could serve as a therapeutic target to minimize vascular complications in diabetes.

## Figures and Tables

**Figure 1 cells-13-00236-f001:**
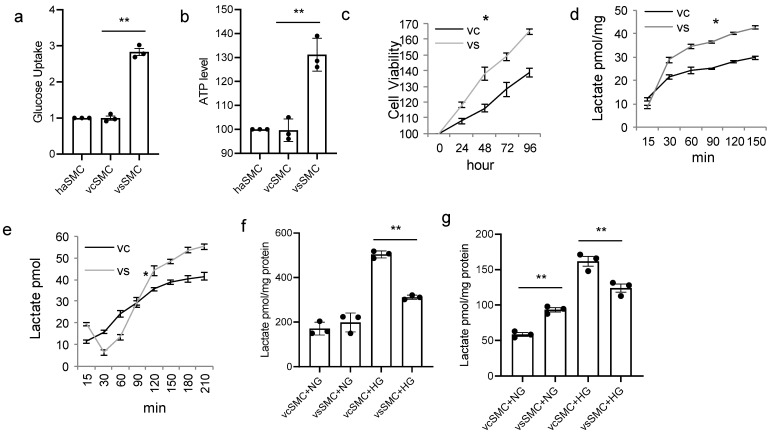
Contractile vSMCs and synthetic vSMCs respond to high glucose concentration differently. (**a**) Glucose uptake in smooth muscle cells (SMCs). Contractile vSMCs (vcSMCs), synthetic vSMCs (vsSMCs), and human arterial SMCs (haSMCs) (Fisher Scientific, USA) were cultured in basic DMEM with 25 mM glucose for 20 min; then, intracellular glucose concentrations were determined (Abcam, Cambridge, MA, USA) and normalized to measurements in haSMC. (**b**) ATP production in SMCs. haSMCs and vSMCs were cultured in 25 mM glucose; then, intracellular ATP levels were measured at 24 h, normalized to the level of haSMCs, and presented as a percentage. (**c**) Cell viability of SMC. Contractile vSMCs and synthetic vSMCs show different viability in high-glucose medium. (**d**,**e**) Dynamic change in lactate in glucose culture. The vSMCs derived from hiPSCs were cultured in (**d**) 5.5 mM (normal glucose, 1 g/L, NG) and (**e**) 25 mM glucose (high glucose, 4.5 g/L, HG). Lactate was measured at indicated time points with lactate assay kit (Biovision, Milpitas, CA, USA). Contractile vSMCs and synthetic vSMCs produce lactate differently depending on glucose concentration. (**f**,**g**) Comparison of lactate production in medium of contractile vSMC and synthetic vSMC culture (**f**) and in cell lysate of contractile vSMCs and synthetic vSMCs cultured in NG and HG (**g**). vSMCs produce more lactate in high-glucose culture. Contractile vSMCs release more lactate into the medium and have more lactate in the cell lysate than synthetic vSMCs. All data are shown as mean ± S.E. * *p* < 0.05, ** *p* < 0.01, for all panels; n ≥ 3 independent experiments for each experimental condition or group.

**Figure 2 cells-13-00236-f002:**
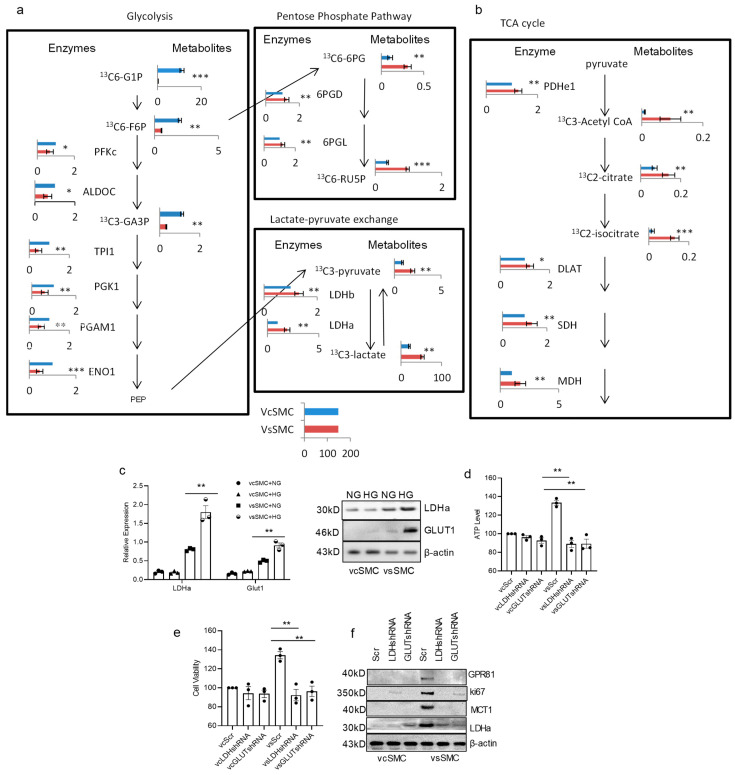
vSMCs utilize glucose differently. There are distinct metabolic differences between contractile vSMCs and synthetic vSMCs in proteomics and metabolomics analyses. Metabolic pathway map summarizing only the significant results from proteomics and [^13^C]-labeled glucose metabolomics analyses. Cell sample treatment and handling were described in the method section. The bar graphs represent the detected levels of proteins or [^13^C]-labeled metabolites in synthetic vSMCs (red bar) and contractile vSMCs (blue bar). All data were obtained from independent experiments. Data present 30 min ^13^C pulse samples. (**a**) Glycolysis pathway (left panel) and lactate production (right panel) map summarizing the results from proteomics and [^13^C]-labeled glucose metabolomics analyses. Glycolysis activity in synthetic vSMCs was lower than contractile vSMCs but lactate production was higher in synthetic vSMCs. (**b**) The TCA cycle pathway map summarizing the results from proteomics and [^13^C]-labeled glucose metabolomics analyses. The higher enzymes and metabolites indicate that the TCA cycle activity in synthetic vSMCs was higher than in contractile vSMCs. (**c**) The expression of glucose transporter 1 (*GLUT1*) and lactate dehydrogenase a (*LDHa*) in vSMCs in NG and HG culture was evaluated in RT-PCR and Western blot analysis. vSMCs were transduced with *LDHa*,* GLUT1*, and scrambled shRNAs and culture for 48 h. Then, the cells were used to measure ATP production (**d**), proliferation with MTT kit (Roche, USA) (**e**), and expression of GLUT1, MCTs, and LDHa with Western blot (**f**) in vcSMCs and vsSMCs cultured for 24 h in HG culture medium. Knocking down LDHa and GLUT1 inhibited the ATP production, cell proliferation, and increase in MCT1, GPR81, and Ki67 in vSMCs in HG culture. All data are shown as mean ± S.E. * *p* < 0.05, ** *p* < 0.01, *** *p* < 0.001 for all panels.

**Figure 3 cells-13-00236-f003:**
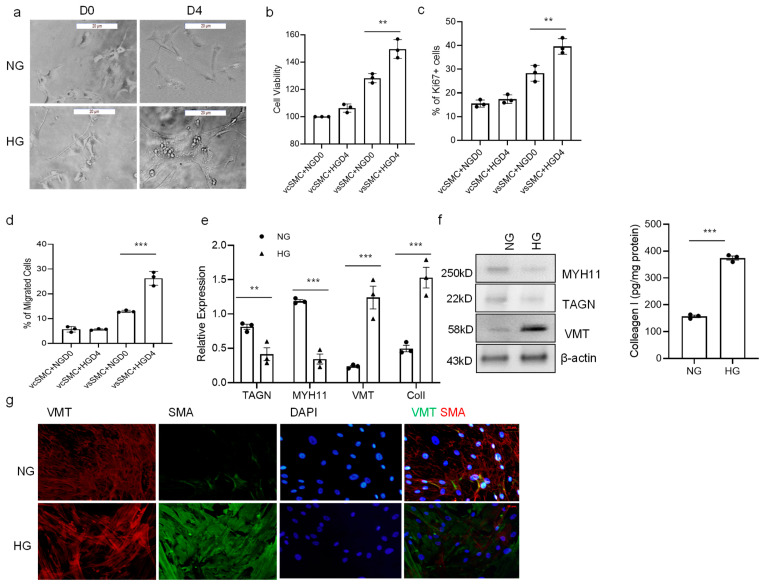
High glucose concentration promotes SMC phenotype switch. (**a**) The contractile vSMCs were cultured in 5.5 mM (NG, normal glucose concentration) and 25 mM glucose (HG, high glucose concentration) for 4 days, and their morphologies were evaluated via light microscopy; bar = 100 μm. D0: day 0; D4: day 4. (**b**) vSMC proliferation was measured with MTT assay (Millipore, St Louis, MO, USA) at days 0 and 4 after vSMCs were cultured in NG and HG for 4 days. (**c**) Ki67 immunofluorescence stain. vSMCs were fixed, stained with ki67 antibody and DAPI, and visualized with Leica microscope cells. (**d**) vSMC migration was measured with migration kit (Millipore, St Louis, MO, USA) at day 4 after vSMCs were cultured in NG and HG for 4 days. (**e**) The expression of markers for the contractile (myosin heavy chain 11 [*MYH11*], transgelin (*TAGN*)) and synthetic (collagen I [*Col 1*] and vimentin [*VMT*]) vSMC phenotype was evaluated via quantitative RT-PCR on day 4 for vSMCs. (**f**) Protein levels of MYH11, TAGN, VMT, and Col I were evaluated with Western blot and ELISA (collagen I, R&D scientific, Minneapolis, MN, USA) on day 4 for vSMCs. (**g**) Immunocytometry. Contractile vSMCs were cultured in NG and HG media for 4 days, the cells were fixed and stained with antibodies against vimentin (VMT), smooth muscle actin (SMA), and DAPI (a nuclear marker), and the images, including overlay (far right), were visualized with an Olympus microscope. Scale bar = 20 μm. There is greater VMT expression in HG-cultured vSMC. All data are shown as mean ± S.E. ** *p* < 0.01, *** *p* < 0.001 for all panels; n ≥ 3 independent experiments for each experimental condition or group.

**Figure 4 cells-13-00236-f004:**
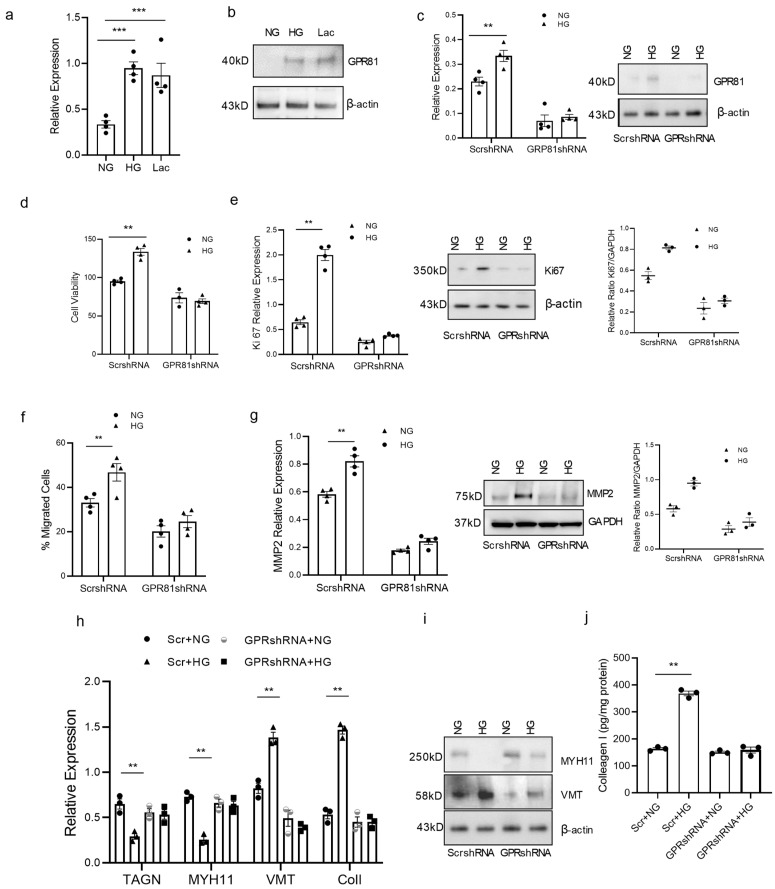
GPR81 expression in vSMC varies with glucose concentration and is required for vSMC ECM production and migration.Contractile vSMC were cultured in normal glucose concentration (NG, 5.5 mM), in high glucose concentration (HG, 25 mM) or in lactate (10 mM) media for 4 days; then, the cells were used to evaluate GPR81 in (**a**) RT-PCR (**b**), Western blot. GPR81 was expressed higher in contractile vSMC cells cultured in HG and lactate. (**c**) Knocking down GPR81 with GPR81shRNA shows the GPR81 expression was decreased in both NG and HG conditions, a finding confirmed with RT-PCR and Western Blot. (**d**). Contractile vSMC with scramble shRNA have higher proliferation in HG than NG culture, but inhibited proliferation in vSMC with GPR81 shRNA in HG culture. (**e**) Contractile vSMC with scramble shRNA have higher proliferation marker ki67 in HG than NG culture but inhibited proliferation in vSMC with GPR81 shRNA in HG culture with RT-PCR and Western blot. (**f**) Contractile vSMC with scramble shRNA have higher migration in HG had higher proliferation than NG culture but inhibited migration in vSMCs with GPR81 shRNA in HG culture. (**g**) Contractile vSMC with scramble shRNA have higher MMP2 in HG than NG culture but inhibited in vSMC with GPR81 shRNA in HG culture with RT-PCR and Western blot. The expression levels of synthetic vSMC marker vimentin (VMT), contractile markers transgelin (TAGN) and myosin heavy chain 11(MYH11) and ECM protein collagen I in same set cell lysate as d and f were evaluated via quantitative RT-PCR (**h**), Western blot (**i**) and collagen I levels were determined via ELISA (**j**). All data are shown as mean ± S.E. ** *p* < 0.01, *** *p* < 0.001 for all panels; n ≥ 3 independent experiments for each experimental condition or group.

**Figure 5 cells-13-00236-f005:**
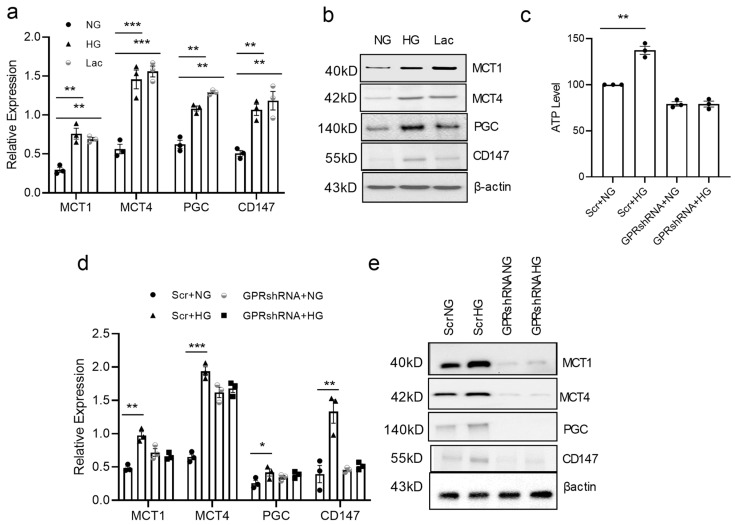
GPR81 regulates expression of genes involved in lactate metabolism. Contractile vSMCs were cultured in fresh DMEM containing NG (5.5 mM glucose), HG (25 mM glucose), or lactate (10 mM) for 4 days and the cells were used to analyze *MCT1*,* MCT4*,* PGC-1α*, and *CD147* mRNA levels with RT-PCR (**a**) and protein levels with Western blot (**b**). MCTs, PGC-1α, and CD147 are higher in HG or lactate culture than in NG culture. Relative mRNA expression via RT-PCR (**c**) and protein by Western blot (**d**). Knocking down GPR81 inhibited the increase in RNA levels of *MCT1*,* MCT4*,* PGC-1α*, and *CD147* with RT-PCR and Protein levels of MCT1, MCT4, PGC-1α, and CD147 with Western blot analysis (**e**). All data are shown as mean ± S.E. * *p* < 0.05, ** *p* < 0.01, *** *p* < 0.001 for all panels; n ≥ 3 independent experiments for each experimental condition or group.

**Figure 6 cells-13-00236-f006:**
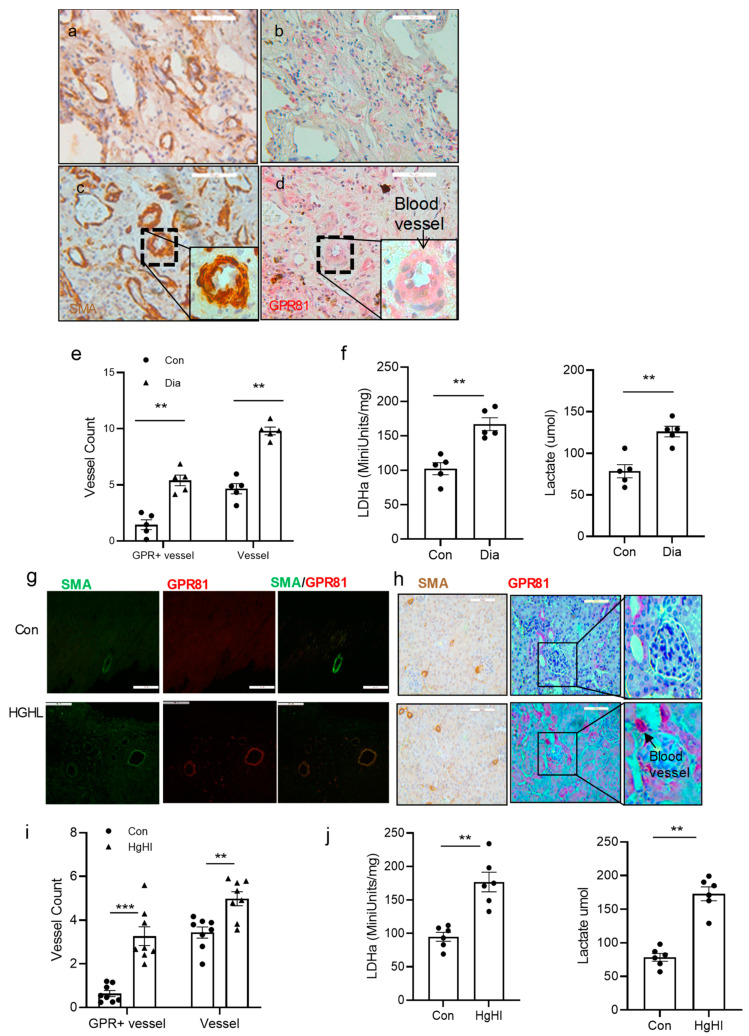
GPR81 in diabetic patients and high-lipid, high-glucose diet TallyHo mice. Tissues from diabetic patients were used to analyze GPR81 expression in vSMCs and surrounding cells with immunohistochemistry (IHC). Serial 5 µm sections of human lung (**a**–**d**) were used for IHC. Diabetic lung tissues had higher blood vessel counts (SMA staining in brown, quantified with Image J Program) and GPR81 expression (red) than control tissues. (**a**) Control nondiabetic lung tissue stained with SMA; (**b**) GPR81 stain on serial section close to section (**a**); (**c**) diabetic lung tissue stained with SMA; (**d**) GPR81 stain on serial section close to section (**c**). (**e**) The blood vessels were counted in the above IHC slides, and the result showed that there are more blood vessels and GPR81-positive blood vessels in diabetic tissues (n = 5) than in control tissues (n = 5). A total of 4 fields (200X) from each case was counted. (**f**) Lung tissue lactate assay. Human lung tissue from control and diabetic patients were used to measure lactate. Diabetic lung demonstrated higher LDHa activity (left) and lactate levels (right) than the control sample. (n = 5). TallyHo mice were also used to observe if the high-glucose diet affects the blood vessels. (**g**,**h**) Tissues from TallyHo mice were used to analyze GPR81 expression in the SMCs and cells around them in immunohistochemistry analysis. (**g**) High-lipid, high-glucose diet promoted GPR81 expression in mouse heart tissue blood vessel SMCs. SMA staining in green and GPR81 staining in red. Scale bar = 50 µm. (**h**) High-lipid, high-glucose diet promoted GPR81 expression in mouse kidney tissue blood vessel SMCs. SMA staining in brown (left panel) and GPR81 staining in red. Scale bar = 100 µm. Far-right panel: enlarged kidney image (glomerulus) with blood vessel labeled. (**i**) The blood vessels were counted in the above IHC slides, and the result showed that there were more GPR81-positive cells in blood vessels of high-lipid, high-glucose diet mouse tissue (n = 8) than normal diet (n = 8). The Image J program was used in quantification of fluorescent density, and 4 fields (400X) were counted in each case. (**j**) Tissue lactate assay. TallyHo mouse heart tissue from control and high-fat, high-glucose TallyHo mice were used to measure LDH activity (left) and lactate levels (right). High-lipid, high-glucose TallyHo mice presented higher lactate levels than control group. (n = 6). Normal diet: Con; high-lipid, high-glucose diet: HGHL. All data are shown as mean ± S.E. ** *p* < 0.01, *** *p* < 0.001 for all panels; n ≥ 3 independent experiments for each experimental condition or group.

**Table 1 cells-13-00236-t001:** List of RT-PCR primers.

Name	Forward	Reverse	Gene No.
*GPR81*	GCCTGCCTTTTCGGACAGACTA	ACCACCGTAAGGAACACGATGC	ENSG00000196917
*Glut1*	TGTGCAACCCATGAGCTAA	CCTGGTCTCATCTGGATTCT	ENSG00000117394
*MCT1*	CTCTGGGCGCCGCGAGATAC	CAACTACCACCGCCCAGCCC	ENSG00000155380
*MCT4*	CCAGGCCCACGGCAGGTTTC	GCCACCGTAGTC ACTGGCCG	ENSG00000168679
*TAGLN*	GAAGAAAGCCCAGGAGCATAA	CCAGGATGAGAGGAACAGTAGA	ENSG00000149591
*MYH11*	AGGCGAACCTAGACAAGAATAAG	CTGGATGTTGAGAGTGGAGATG	ENSG00000133392
*LDHa*	AGCCCGATTCCGTTACCT	CACCAGCAACATTCATTCCA	ENSG00000134333
*Vimentin*	TCGTTTCGAGGTTTTCGCGTTAGAGAC	GACTAAAACTCGACCGACTCGCGA	ENSG0000026025
*PGC1a*	AGCCTCTTTGCCCAGATCTT	GGCAATCCGTCTTCATCCAC	ENSG00000109819
*CD147*	ACCTGCTCTCGGAGCCGTTCA	CGTCTCTGCACAGATTGCAT	ENSG00000172270

## Data Availability

All data are available from the corresponding author upon reasonable request.

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
