# Peer review of "High Glucose Levels Promote Switch to Synthetic Vascular Smooth Muscle Cells via Lactate/GPR81"

_cells, 2024, doi:10.3390/cells13030236_

Round 1
Reviewer 1 Report (Previous Reviewer 2)
Comments and Suggestions for Authors
The authors have partially addressed problems from the previous version. Notably, in Figure 6 they have included lung and kidney sections and removed the problematic liver images (although these are still mentioned in the text). These Figure 6 data are an improvement.
The authors apparently have no molecular weight markers or controls to support the specificity their western analyses, and deem it impractical to 'repeat all experiments.' This is an unfortunate situation.
Comments on the Quality of English LanguageThe manuscript would benefit from close editing to improve the flow and clarity of the English language. This includes the figure legends.
Author Response
The authors have partially addressed problems from the previous version. Notably, in Figure 6 they have included lung and kidney sections and removed the problematic liver images (although these are still mentioned in the text). These Figure 6 data are an improvement.
Response:
Thank you so much for your comments. We took away our comment about liver IHC result.
The authors apparently have no molecular weight markers or controls to support the specificity their western analyses, and deem it impractical to 'repeat all experiments.' This is an unfortunate situation.
Response:
Thank you so much for your comments. We had marked the molecular weight on all Western blot band in the figures.
Western blot analysis practice as following at our lab: after proteins been transferred to blot, the blot would be cut into multiple piece for different targeting proteins (reacting to different antibodies) in most cases. Specificity relys on sample arrangement and antibody. it always is repeatable.
Comments on the Quality of English Language
The manuscript would benefit from close editing to improve the flow and clarity of the English language. This includes the figure legends.
Response:
Thank you so much for your suggestion. We have double checked the manuscript carefully.
Reviewer 2 Report (Previous Reviewer 3)
Comments and Suggestions for Authors
Please fix the supplemetory material according to my suggestions of previous reviews.
Author Response
we reorganized the images and saved in "uncroppedimage" and will be uploaded as supplement file. Please note that we do not have big blot because of the way we do our Western blot analysis at our lab. After proteins been transferred to blot, the blot would be cut into multiple pieces for different targeting proteins (reacting to different antibodies) in most cases. the blot is very small to save the antibody.
This manuscript is a resubmission of an earlier submission. The following is a list of the peer review reports and author responses from that submission.
Round 1
Reviewer 1 Report
Comments and Suggestions for Authors
The authors showed that vSMCs became more synthetic phenotype when cultured in high glucose medium via lactate/GPR81 pathway. The hypothesis is quite interesting, however, the quality of the manuscript needs to be much improved.
Overall, the manuscript is not well organized. Introduction did not support and provide sufficient information for the study. The methods section does not provide enough information to understand the results. (The result section should be a little clearer. Background explanations should be provided in the introduction and the discussion.) I suggest rewriting the manuscript and resubmitting it to journal.
minor
Abbreviations should not be in the first place in sentences. (E.g. vSMC in line 89, etc)
Many abbreviations in the manuscript are confusing. (e.g. vcSMC, vsSMC, etc. It is better use contractile vSMC and synthetic vSMC instead of using vcSMC, vsSMC.
Comments on the Quality of English Language
Extensive editing of English language required.
Author Response
Reviewer 1:
Comment:
I suggest rewriting the manuscript and resubmitting it to the journal.
Minor
Abbreviations should not be placed at the beginning of sentences (e.g., vSMC in line 89, etc). Many abbreviations in the manuscript are confusing (e.g., vcSMC, vsSMC, etc). It is better to use "contractile vSMC" and "synthetic vSMC" instead of using "vcSMC" and "vsSMC."
Response:
Thank you so much for your suggestion. We have rewritten most parts of the manuscript, and two of the authors, who are native English speakers, have carefully edited the manuscript. All instances of vcSMC have been changed to contractile vSMC, and vsSMC has been changed to synthetic vSMC.

Reviewer 2 Report
Comments and Suggestions for Authors
This manuscript from Yang et al describes studies of hyperglycemic effects on induced pluripotent stem cells (iPSCs) that have been differentiated to model contractile (c) or synthetic (s) vascular smooth muscle cells (SMCs), along with studies of Tallyho mice (a model of type II diabetes) and analysis of liver tissue from diabetic humans.
In brief, the SMCs are reported to respond differently to high glucose, with sSMCs showing higher glucose uptake, ATP production, viability, and variable production of lactate. This differential utilization of glucose is reinforced by metabolomic studies that show sSMCs perform lower levels of glycolysis, and higher conversion of pyruvate to lactate and to TCA cycle intermediates; knockdown studies implicate increased expression of the glucose transporter Glut1 and LDH in these changes. These metabolic changes are then correlated with further effects of high glucose on sSMC (but not cSMC) migration and dedifferentiation.
The authors extend some of these effects to exogenous lactate and induction of the lactate sensor GPR81, showing that knockdown of this protein prevents high glucose effects, including increased expression of MMP2 and type I collagen; several additional potential GPR81 downstream target genes are identified, including MCT1, MCT4, CD147, and PGC1.
Lastly, the study turns to evaluation of tissues from diabetic mice and humans; these data include stains of human liver and mouse heart.
The outline sketched by this study is interesting, but there are several limitations. Some of these can be addressed by modifying the text, and some may require additional experimentation. Overall, the text needs close editing for proper English.
General points
1. Antibody specificity is not demonstrated. Some supplemental figures provide somewhat larger pictures of western blots (though not full, uncropped blots), but these provide no molecular weight marks and do not confirm the specificity of signals shown throughout the dataset.
2. The claim that differentiated iPSCs perform as models of stable synthetic vs contractile SMCs is difficult to understand here, since a central point of the study is to show that the phenotypes change in response to high glucose. Perhaps the stability of the phenotype could be better demonstrated, or the authors could clarify their meaning.
3. The data presented should be described more clearly. For several panels, including Fig. 1, for example, the legend reports “n≥3 independent experiments for each experimental condition or group”. Do the figures presented show 3 independent biological replicates, or technical replicates performed together? The glucose concentrations should be clarified – high glucose appears to refer to either 22 or 25 mM. Why is this different?
Specific points
For Fig. 2, the total glucose concentration relevant for the metabolomics study is unclear. Is this study reflective of normal or high glucose conditions, which have different effects on glucose handling?
Fig. 3 legend states “High glucose concentration promotes SMC phenotype switch via lactate”. The data in this figure do not implicate lactate.
Fig. 4 shows that GPR81 knockdown appears to decrease MMP2 qRT-PCR levels regardless of glucose concentration. How is this explained? Is MMP2 activity affected by these conditions?
Fig. 6. The tissue studies are difficult to interpret, and should be improved. It is unclear why human liver and mouse heart have been chosen. For the human tissues, background staining appears high. Although serial sections are reported for liver, these sections do not appear to be close, and tissues appear distorted. The liver section appears highly fibrotic. The immunofluorescent stain of the mouse heart is not evident in the current figure.
Comments on the Quality of English Language
The manuscript needs close editing to correct the English. There are some problems with word choice, e.g., 'feathers'.
Author Response
Comment 1:
Antibody specificity is not demonstrated. Some supplemental figures provide somewhat larger pictures of western blots (though not full, uncropped blots), but these do not include molecular weight markers and do not confirm the specificity of the signals shown throughout the dataset.
Response:
We have been collecting western blot images without molecular weight markers, following instructions from the Journal of Clinical Investigation years ago. Additionally, we had to cut the blots into small pieces for different antibodies. Please refer to the uncut image data.
Comment 2:
The claim that differentiated iPSCs perform as models of stable synthetic vs. contractile SMCs is difficult to understand here, as a central point of the study is to show that the phenotypes change in response to high glucose. Perhaps the stability of the phenotype could be better demonstrated, or the authors could clarify their meaning.
Response:
Primary smooth muscle cell lines consist of a mixture of synthetic and contractile SMCs, making it very difficult to observe differences between these two phenotypes. Since previous publications have addressed these issues, we cut off the statement in this manuscript.
Comment 3:
The data presented should be described more clearly. For several panels, including Fig. 1, the legend reports "n≥3 independent experiments for each experimental condition or group." Do the figures presented show 3 independent biological replicates or technical replicates performed together? The glucose concentrations should be clarified - high glucose appears to refer to either 22 or 25 mM. Why is this different?
Response:
We apologize for the oversight. The data is from 3 independent biological replicates, and each data point is the average of 3 technical replicates. The glucose concentration has been corrected in the manuscript.
Specific points
For Fig. 2, the total glucose concentration relevant for the metabolomics study is unclear. Is this study reflective of normal or high glucose conditions, which have different effects on glucose handling?
Response:
We pulsed 13C-glucose in both normal and high glucose conditions at 15, 30, 60, and 120 minutes. No difference was found between phenotypes in normal glucose. Most differences in metabolites were found between phenotypes at 30 and 60 minutes. The presented data are from the 30-minute time point.
Fig. 3 legend states "High glucose concentration promotes SMC phenotype switch via lactate." The data in this figure do not implicate lactate.
Response:
We apologize for the oversight. The legend has been corrected.
Fig. 4 shows that GPR81 knockdown appears to decrease MMP2 qRT-PCR levels regardless of glucose concentration. How is this explained? Is MMP2 activity affected by these conditions?
Response:
GPR81 may be closer to MMP2 in the signal transduction pathway in this experiment.
Fig. 6:
The tissue studies are difficult to interpret and should be improved. It is unclear why human liver and mouse heart have been chosen. For the human tissues, background staining appears high. Although serial sections are reported for the liver, these sections do not appear to be close, and the tissues appear distorted. The liver section appears highly fibrotic. The immunofluorescent stain of the mouse heart is not evident in the current figure.
Response:
We only had access to those tissues, and we used them to demonstrate if there is GPR81 expression. Most sections appeared quite fibrotic in our experiment, which may be due to the fact that most of the tissue was collected from aged veterans who often had alcohol problems. We had observed diabetes lung tissue sections that presented a similar pattern to the liver. We thought blood vessels in any organ should be fine for this purpose, as diabetes affects most blood vessels in important organs. We apologize for our poor technique; we were unable to improve it within the 10-day timeframe.

Reviewer 3 Report
Comments and Suggestions for Authors
The authors investigated the effects of normal and high glucose concentrations on lactate formation and phenotypic traits of synthethetic and contractile VSMCs. There is a number different experiments and the paper is not always easy to follow. The main conclusion is that that high glucose concentration promotes synthetic vSMC phenotype via lactate/GPR81.
Specific comments:
Line 80: It is suggested to rephrase this. Lactate is by no means a waste product but a substrate for gluconeogenesis and an energy source for cardiac and type I skeletal muscle fibers.
Line 92: Is it possible to formulate a hypothesis that was driving this study?
Line 109: Where are the hypoxia experiments?
Line 158 and below: Please provide the primer sequences as table including gene bank accession numbers.
Line 187: Online Table 1 was missing. Please provide.
General comment on the figures: The quality of the graphic presentation is low, alignment of panels, size of panels etc. Please improve this.
Figure 1C: Please explain the meaning of the y-axis on cell viability.
Figure 1D: pmol/mg protein?
Figure 1E: pmol/what?
Line 305: Release or production? It is produced in the cells.
Lines 306-7: What is the eveidence for the notion that vsSMCs produce more lactate than vcSMCs if lactate is higher in both medium and the lysate of vcSMCs? Panel f and g.
Line 317: 14C or 13C? See line 231. Please correct including respective figures and legends.
Figure 2e: Please explain the meaning of a variabilty of approximately 130, see comment on Figure 1C.
Figure 3g: There should be a clear SMA signal under normal glucose conditions. In the overlay there is hardly an SMA or VMT signal. Please check if the color coding (green vs. red) is correct in the legand above the overlay.
Fig 6g: Signals hard hard to see and no differences can be observed.
Fig 6 h and i: Please check the correctness and meaning of the units of the y axes.
Please revise lines 529-530 to provide clear information. Why was cardiac tissue investigated if the focus of the study was on vSMCs?
Original images: Please provide full Western blot images that are in a clear order without overlaps.
Comments on the Quality of English LanguageThere are several issues regarding language that need attention:
Line 111: specimens were obtained...
Line 195: cells were cultured...
Lines 269-70: Please remove the comment in parentheses.
Line 419: investigated
Line 486: control rather than sham-group in the given context
Line 546: uptake
Author Response
Comments and responses:
Line 80:
It is suggested to rephrase this. Lactate is by no means a waste product but a substrate for gluconeogenesis and an energy source for cardiac and type I skeletal muscle fibers.
Response:
Thank you for the suggestion.
Line 92:
Is it possible to formulate a hypothesis that was driving this study?
Response:
We formed a hypothesis later in that paragraph.
Line 109:
Where are the hypoxia experiments?
Response:
We apologize for the oversight. This has been corrected.
Line 158 and below:
Please provide the primer sequences as a table, including gene bank accession numbers.
Response:
This has been corrected.
Line 187:
Online Table 1 was missing. Please provide.
Response:
We will add it to the submission.
General comment on the figures:
The quality of the graphic presentation is low, with alignment of panels and the size of panels needing improvement.
Figure 1C:
Please explain the meaning of the y-axis on cell viability.
Response:
We used the term as it is presented in the product instructions. We assume it means the number of healthy or proliferating cells.
Figure 1D:
pmol/mg protein?
Response:
The data has been normalized with the protein concentration in cell lysate.
Figure 1E:
pmol/what?
Response:
This has been corrected.
Line 305:
Release or production? It is produced in the cells.
Response:
This has been corrected.
Lines 306-7:
What is the evidence for the notion that vsSMCs produce more lactate than vcSMCs if lactate is higher in both medium and the lysate of vcSMCs? Panels f and g.
Response:
This has been corrected.
Line 317:
14C or 13C? See line 231. Please correct, including respective figures and legends.
Response:
This has been corrected.
Figure 2e:
Please explain the meaning of a variability of approximately 130, as mentioned in the comment on Figure 1C.
Response:
We used the term as it is presented in the product instructions. We assume it means the number of healthy or proliferating cells.
Figure 3g:
There should be a clear SMA signal under normal glucose conditions. In the overlay, there is hardly an SMA or VMT signal. Please check if the color coding (green vs. red) is correct in the legend above the overlay.
Response:
This has been corrected.
Fig 6g:
Signals are hard to see, and no differences can be observed.
Response:
This has been corrected.
Fig 6 h and i:
Please check the correctness and meaning of the units of the y-axes.
Response:
This has been improved.
Lines 529-530:
Please revise to provide clear information. Why was cardiac tissue investigated if the focus of the study was on vSMCs?
Response:
We thought blood vessels in any organ should be fine for this purpose, as diabetes affects most blood vessels in important organs. We did analyze other tissues, but the background staining was rather high.
Original images:
Please provide full Western blot images that are in a clear order without overlaps.
Response:
This has been corrected.
There are several language issues that need attention:
Line 111:
Specimens were obtained...
Response:
This has been corrected.
Line 195:
Cells were cultured...
Response:
This has been corrected.
Lines 269-270:
Please remove the comment in parentheses.
Response:
This has been corrected.
Line 419:
Investigated
Response:
This has been corrected.
Line 486:
Control rather than sham group in the given context.
Response:
This has been corrected.
Line 546:
Uptake
Response:
This has been corrected.

Round 2
Reviewer 2 Report
Comments and Suggestions for Authors
The authors have provided marginal responses to my prior critique.
1. Western blotting - with no molecular weight markers and no controls, the specificity of the signals provided remains uncertain
2. The best way to validate the synthetic vs contractile SMC phenotypes studied in cell culture is by confirming in the in vivo setting. Unfortunately, the samples in Figure 6 are quite limited for this (see below).
3. regarding the metabolomics study (Fig. 2), the answer provided is confusing, so it will be difficult for others to replicate this experiment. It sounds like the study shown is with high glucose at 30 minutes – this should be clearly stated in the legend.
4. "GPR81 may be closer to MMP2 in the signal transduction pathway in this experiment."
This statement does not explain much – please clarify. Also, the claim that MMP2 is a 'marker of migration' is potentially misleading
Fig. 6 images of diseased liver and heart remain problematic. These tissues selected for these studies are not well considered. Vascular diseases of the liver are rare, and macrovascular disease such as atherosclerosis (the key problem for diabetics) is not generally seen (e.g., see PMID 19399912). The signals in the mouse heart vessels are barely visible, which undermines the data quality.
Comments on the Quality of English LanguageThe quality of the English language is better.
Reviewer 3 Report
Comments and Suggestions for Authors
The manuscript improved. The file containig original images for blots and gels should provide full Western blots as asked for in my first review. Furthermore, the material should be arranged more orderly.
Comments on the Quality of English Language
The added text (in red) should be checked for grammar again.